# So far, so good… Similar fitness consequences and overall energetic costs for short and long-distance migrants in a seabird

**David Pelletier** [1,2]⊛ *, **Yannick Seyer** [3]⊛, **Stefan Garthe**[4‡], **Salomé Bonnefoi**[2‡], **Richard A. Phillips**[5‡], **Magella Guillemette**[2]⊛

**1** Département de biologie, Cégep de Rimouski, Rimouski, Québec, Canada, **2** Département de biologie, Université du Québec à Rimouski, Rimouski, Québec, Canada, **3** Département de biologie and Centre d'études nordiques, Université Laval, Québec, Québec, Canada, **4** Research and Technology Centre (FTZ), University of Kiel, Buesum, Germany, **5** British Antarctic Survey, Natural Environment Research Council, Cambridge, United Kingdom

⊛ These authors contributed equally to this work.
‡ These authors also contributed equally to this work.
* david.pelletier@cegep-rimouski.qc.ca

**Data Availability Statement:** All data are available in the Supporting Information file.

## Abstract

Although there is a consensus about the evolutionary drivers of animal migration, considerable work is necessary to identify the mechanisms that underlie the great variety of strategies observed in nature. The study of differential migration offers unique opportunities to identify such mechanisms and allows comparisons of the costs and benefits of migration. The purpose of this study was to compare the characteristics of short and long-distance migrations, and fitness consequences, in a long-lived seabird species. We combined demographic monitoring (survival, phenology, hatching success) of 58 Northern Gannets (*Morus bassanus*) breeding on Bonaventure Island (Canada) and biologging technology (Global Location Sensor or GLS loggers) to estimate activity and energy budgets during the non-breeding period for three different migration strategies: to the Gulf of Mexico (GM), southeast (SE) or northeast (NE) Atlantic coast of the U.S. Survival, timing of arrival at the colony and hatching success are similar for short (NE, SE) and long-distance (GM) migrants. Despite similar fitness consequences, we found, as expected, that the overall energetic cost of migration is higher for long-distance migrants, although the daily cost during migration was similar between strategies. In contrast, daily maintenance and thermoregulation costs were lower for GM migrants in winter, where sea-surface temperature of the GM is 4-7° C warmer than SE and NE. In addition, GM migrants tend to fly 30 min less per day in their wintering area than other migrants. Considering lower foraging effort and lower thermoregulation costs during winter for long-distance migrants, this suggests that the energetic benefits during the winter of foraging in the GM outweigh any negative consequences of the longer-distance migration. These results support the notion that the costs and benefits of short and long-distance migration is broadly equal on an annual basis, *i.e.* there are no apparent carry-over effects in this long-lived bird species, probably because of the favourable conditions in the furthest wintering area.

**Funding:** Natural Sciences and Engineering Research Council of Canada (NSERC) Discovery Grant (to MG) and Fonds de recherche du Québec Nature et Technologies (FRQNT) – Programme de recherche pour les chercheurs de collège (to DP). The funders had no role in study design, data collection and analysis, decision to publish, or preparation of the manuscript.

**Competing interests:** The authors have declared that no competing interests exist.

## Introduction

Migration has a large impact on the distribution of birds around the globe. Each year, under the pressure of cyclic environmental conditions, it is estimated that 1,933 of the world's 10,761 extant bird species migrate between their breeding range and wintering quarters [1]. The ultimate factors that drive the evolution of migration in birds were proposed a long time ago, and include seasonality in food availability [2, 3], climatic conditions [4] and predation [5–7]. Although migration must be linked to some advantages on an evolutionary time-scale, the proximate factors and mechanisms that underlie this seemingly costly behaviour remain difficult to identify. For instance, although the energetic cost of locomotion is an inherent feature of migration, very often it is estimated crudely from the distance travelled without any information on time spent flying, which is greatly influenced by wind direction and speed [8, 9]. In addition, one hypothesized benefit of migration, reduced thermoregulatory costs on the wintering grounds, is seldom quantified [10, 11].

Why some individuals of the same species migrate further while others stay close to their breeding site is still the central question of studies of migration strategies [10, 12, 13]. Differential migration, broadly defined as observed variation in distance and timing among individuals within a population [14], offers an interesting framework to study the adaptive value of migratory behaviour. Population segregation during migration can be determined by genetic control, or can be a condition-dependent response reflecting body reserves, social status or local food abundance during the breeding period [14–16]. Differential migration also offers unique opportunities to study carry-over effects, whereby processes in one season have consequences in subsequent seasons [17–20]. Use of different wintering areas can affect body condition or survival in the short term, and reproductive success and phenology in the subsequent breeding period [17, 21, 22]. Therefore, a cost-benefit approach applied to a differential migrant may provide insights into the mechanisms driving the migration strategy of individuals [10, 14] but see Newton [15]

The Northern Gannet (*Morus bassanus*) is a plunge-diving predator that nests in large colonies and has a wide breeding distribution in the North Atlantic. High nest-site fidelity, large size and tolerance of disturbance makes this species a very good model for the study of migration using remote-monitoring devices [10, 23–26]. Collectively, these studies have shown that gannets (a) use oriented migration (as opposed to dispersive migration) to reach alternative wintering quarters used by particular colonies, (b) are highly philopatric to their wintering areas from year to year, and (c) time their departure from breeding and wintering areas depending on distance to their destination. These studies have greatly increased our knowledge of migration tactics and identified distribution hotspots at sea that warrant protection. However, only one has addressed the costs and benefits of alternative migration strategies, comparing the time spent in flight and the energy required for thermoregulation for wintering adults from a colony in western Europe [10], and none compared fitness, behavioural and energetic aspects throughout an annual cycle.

In this paper, we examine alternative migration strategies of gannets breeding in the Gulf of St Lawrence, Canada, tracked using devices (Global Location Sensor or GLS loggers) that record the time spent flying and sea surface temperature (SST), together with light data which can be used to estimate location. Our objectives were to examine evidence for carry-over effects by comparing survival, hatching success and phenology among individuals with different strategies and the costs and benefits of short and long-distance migration in terms of thermoregulation, and time and energy spent in flight to and within the wintering areas. The combination of these data allow us to test one of the basic principles of differential migration,

that there is no long-term cost to longer migration if the distant wintering area provides better conditions.

# Materials and methods

## Study site and fieldwork

The fieldwork was conducted on Bonaventure Island (48˚30' N, 64˚09' W) located in the Île-Bonaventure-et-du-Rocher-Percé National Park in the Gulf of St. Lawrence (Quebec, Canada) where a long-term study started in 2008. Annually, this colony is monitored for survival, breeding phenology and hatching success. Chick-rearing adult Northern Gannets were captured using a noose-pole within four plots in the peripheral section of the colony, prior to the fall migration. Bird capture and handling methods were approved by the Animal Care Committee (ACC) of the Université du Québec à Rimouski, and complied with the guidelines of the Canadian Council on Animal Care (CCAC). These birds were marked with a U.S. Fish and Wildlife Service steel ring and a color-coded plastic band, and equipped with a Global Location Sensors (Mk15 GLS loggers; British Antarctic Survey, Cambridge, United Kingdom).

## Global Location Sensor (or GLS) loggers

The GLS loggers had three types of sensors (light, temperature and immersion) and were attached using cable-ties to the steel ring. Fifty-eight and 36 devices were deployed on different individuals in September 2010 and August 2011, respectively. Of the GLS deployed in 2010, 43 were retrieved in May and June 2011, and one in May 2012 (recapture rate = 76%). Of the GLS deployed in 2011, 31 were retrieved in May and June 2012, and four in May and June 2013 (recapture rate = 97%). Of the 79 loggers recovered, 58 provided data for the whole period of deployment and were used for our analyses; the remainder failed at some point because of battery discharge, water ingress or malfunction of one of the sensors.

The GLS loggers weighed 2.5 g (<0.1% of adult body mass; mean ± SE of 3,030 ± 302 g) and were 16 x 14 x 6 mm. The loggers measured light levels every minute and record the maximal values every 10 min (data used for geolocation). In addition, immersion in saltwater (wet/dry) was tested every 3 s and integrated over 10 min intervals, thereby providing a value between 0 and 200 that indicates the proportion of time that birds spent in water or in flight during the non-breeding period when birds were away from land; these were used to construct at-sea activity budgets, as used in many other studies [27–30]. The loggers also recorded water temperature after 20 and 40 min spent continuously wet.

## Data analysis

**Survival.**  Survival analyzes were carried out with the software Mark [31] using the Cormack-Jolly-Seber capture-mark-recapture model [32]. Goodness-of-fit was tested with the software U-Care [33]. We tested for transience (presence of individuals that were temporary visitors and did not return after their first capture) with test 3, and for trap-dependence (the probability that recaptures were not independent between occasions, *i.e.* trap-shyness or trap-happiness) with test 2. Test 3 was separated in two parts: test 3.SR tested the hypothesis that there was no difference in the probability of recapturing an individual that was newly or previously marked, and test 3.SM tested the hypothesis that there was no difference in the expected time of recapture between newly and previously marked individuals. Test 2 was also separated in two parts. Test 2.CT tested the hypothesis that there was no difference in the probability of being recaptured between individuals captured or not captured at the previous occasion. Test 2.CL tested the hypothesis that for individuals not immediately recaptured there was no

**Table 1. Results from the goodness-of-fit test of the Cormack-Joly-Seber (CJS) model comparing effects of migration strategy on survival in Northern Gannets nesting on Bonaventure Island, Canada.** The three wintering destinations were Gulf of Mexico (GM), southeast coast of the U.S. (SE) and northeast coast of the U.S. (NE). The $\chi 2$ value of the CJS model is obtained from the sum of the four components 3.SR, 3.SM, 2.CT and 2.CL.

| Tests | GM migrants | | | SE migrants | | | NE migrants | | |
|---|---|---|---|---|---|---|---|---|---|
| | $\chi^2$ | df | *P*-value | $\chi^2$ | df | *P*-value | $\chi^2$ | df | *P*-value |
| **3.SR** | $1.309 \times 10^{-29}$ | 1 | 1 | 0 | 1 | 1 | $2.429 \times 10^{-28}$ | 1 | 1 |
| **3.SM** | 1.203 | 2 | 0.548 | $2.788 \times 10^{-30}$ | 1 | 1 | $3.425 \times 10^{-28}$ | 2 | 1 |
| **2.CT** | $9.681 \times 10^{-30}$ | 2 | 1 | $4.840 \times 10^{-31}$ | 1 | 1 | 4.911 | 6 | 0.555 |
| **2.CL** | na | na | na | na | na | na | 0 | 2 | 1 |
| **Sum per categories** | 1.203 | 5 | na | 0 | 3 | na | 4.911 | 11 | na |
| **Sum all categories** | 6.114 | 19 | 0.998 | | | | | | |

difference in the expected time of next recapture between the individuals encountered and not encountered at the previous occasion [34]. The overall goodness-of-fit test demonstrated that the model fits well the data ($\chi^2_{19} = 6.114$, $P = 1.00$; **Table 1**). We found no evidence of transience ($z = -1.11 \times 10^{-14}$, 2-sided test, $P = 1.00$) nor trap-dependence ($z = -1.46$, 2-sided test, $P = 0.15$). To test for over-dispersion, we used the variation inflation factor ($\hat{c} = \chi 2 / df$). As we found no over-dispersion ($\hat{c} = 0.322$), we fixed the $\hat{c}$-value to 1. Model selection was done using Akaike's information criterion corrected for small samples (AICc) and AICc weights ($w_i$) [35]. The general model we tested included a survival probability depending on the interaction between time and the category of migrants ($\varphi_{g*t}$) and a recapture probability depending on time only ($p_t$). From this model, we defined 9 other models with either a constant survival probability or a survival probability depending on time, the category of migrants or the addition or interaction of both. The recapture probability was either time dependant or constant throughout time. The 95% confidence intervals were evaluated using the profile likelihood method (also called maximum relative likelihood) [32].

**Phenology and hatching success.** We refer to the time from colony departure in the fall to return in the spring as the "non-breeding period"; the time spent in transit to and from the wintering area as the "fall and spring migration seasons"; and, the time spent in the wintering area as the "winter season". Timing of events during the non-breeding period were determined as above (an example of activity and temperature data used for one individual is illustrated in **Fig 1**). The end of the breeding period (departure date for the fall migration) was defined as the first date with dry time shorter than 12 h per day. Similarly, the onset of the next breeding period (arrival date of the spring migration) was based on a dry time longer than 12 h per day.

Monitoring at the colony was from the end of May, during incubation, until the end of August, when chicks are 3 weeks from fledging. The presence of an egg or chick at each numbered nest was recorded daily. Most chicks died during the summer of 2011 and 2012 as a result of food shortage [36]. To avoid the impact of this problem occurring during the breeding period, we only present data on hatching success and not fledging success. This was calculated for each category of migrants by dividing the number of chicks hatched by the number of eggs laid. We compiled hatching success for each instrumented bird in 2010–2011 and 2011–2012 from 2009 to 2017. Given the very high fidelity to wintering sites in gannets [25], we assume that individuals migrated to the same area each year.

**Geolocation.** Light level data were analyzed using TransEdit2 in the BASTrack software (version 18; British Antarctic Survey, Cambridge) and GeoLight package [37] in R 3.5.3 [38] using established methods [39, 40]. Briefly, sunset and sunrise were estimated using a light-intensity threshold of 16 and a minimum dark period of 4 h. Latitude was derived from day length, and longitude from the timing of local midday and midnight, providing two positions

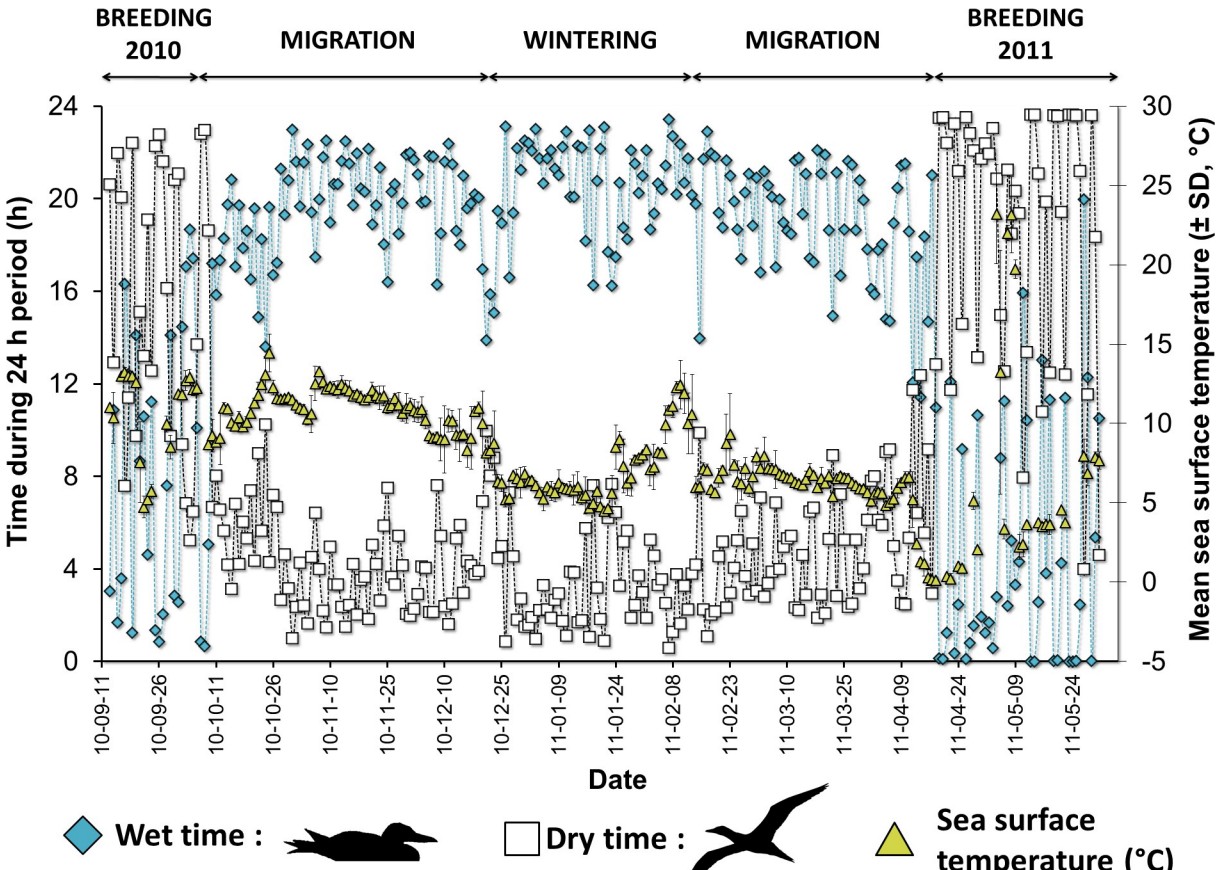

**Fig 1. Example of long-term monitoring of daily activity patterns and temperature recorded using a combined Global Location Sensor (GLS), immersion and temperature logger attached for 260 days on a Northern Gannet nesting on Bonaventure Island and wintering off the northeast coast of the U.S.** Diamonds (♦) depict the *wet time* (h), when the gannet is diving or sitting on the water; squares (□) depict the *dry time* (h), when the gannet is out of the water, flying or on land at the colony; and, triangles (Δ) are the mean sea surface temperature per day (± SD, ˚C). Breeding periods are bounded by dry time per day of more than 12 h, and the date of arrival at the wintering area is based on geolocation using the light data (see *Methods*).

per day with an accuracy of ~ 200 km [39]. The sun elevation angle used for processing of the light data was the value which centred the estimated locations at the colony for each logger during ground-truthing periods before and deployment on the bird. We used a median sun elevation angle calculated for each individual dataset (between -6˚ and -3˚). For each individual, preliminary data selection was according to criteria suggested by Kubetzki *et al.* [23] including the elimination of positions around the equinox [41], those > 200 km inland on the continent (because non-breeding gannets are exclusively pelagic), or associated with an apparent movement greater than the maximum distance (1,200 km) that gannets can travel in a 24 h period (based on the average flight speed of long-distance flight of 58.4 km.h$^{-1}$ [42] and maximum dry time recorded for each individual within the non-breeding period of 19 h). Wintering areas were mapped using kernel density estimation (KDE) [43] in a Lambert conformal conic projection within the Geospatial Modelling Environment software (Version 0.6.0.0) in ArcGIS 10 (ESRI, Canada). Following the standard for many seabird studies, we took the 50% kernel contour to represent regions of high use for each individual [44] and selected a different smoothing parameter (bandwidth or *h*-value) per individual using a likelihood cross-validation algorithm [45]. The total winter range was categorised as one of three regions [46]: 1)

Gulf of Mexico (GM), *i.e.* waters to the west of Florida (east of 81˚W), 2) the southeast coast of the U.S. (SE), from east of Florida to the northern border of South Carolina (south of 34˚N), and 3) northeast coast of the U.S. (NE), between North Carolina and Maine. These zones were chosen on the basis of typical thermal regimes and current systems that describe each zone and that influence prey species communities [25, 47]. We generated a map of centroids of wintering location (Dec. 15th–Feb. 15th, mean ± 95% CI) of each individual in R [38] using the 'ggmap' [48] and 'ggplot2' packages [49] with map tiles by Stamen Design (www.stamen.com) and data by OpenStreetMap, under ODbL, under CC BY 3.0 (creativecommons.org/licenses/by/3.0/) Arrival and departure from the colony were determined from a combination of immersion and geolocation data. Since the spring (return) migration coincided with the spring equinox period (when latitude is impossible to determine from light data using threshold geolocation methods), departure date from the wintering site was estimated from the return date to Bonaventure Island (based on immersion data) assuming the same transit time as on the fall (outward) migration. As observed by Kubetzki *et al.* [23], autumn and spring migrations are consistent in duration for individual gannets. Residence time in the wintering areas was calculated as the interval between arrival after the fall migration, and the estimated departure date for the spring migration.

**At-sea activity (wet/dry) and sea surface temperature.** The immersion data were used to calculate the total time spent wet or dry in each 24 h period during the non-breeding period. Wet time includes all these activities: diving, swimming and resting or sitting on the water surface. Because gannets are exclusively pelagic during the non-breeding period [46], dry time represents flying (indeed, only one study reports two birds using an offshore island as a stopover during spring migration [25]). We used time spent flying per day as a proxy for migration and foraging effort, following Garthe *et al.* [10].

The average (± SD) of all temperature records per day was calculated for each logger, and assumed to represent daily sea surface temperature (SST) experienced by each bird.

**Energetics.** Flight is clearly not a uniform activity because birds can vary flight speed according to prevailing conditions and intention, and switch between flapping and gliding; hence, estimation of flight costs is not straightforward. Pelletier [50] used five different models to estimate flight costs in Common Eiders (*Somateria mollissima*). That study found congruence between empirical measurements and theoretical predictions, when the latter (the aerodynamic model developed by Pennycuick [51] implemented in the software Flight v. 1.24) was corrected for flight muscle efficiency (see below). By comparing the five models, Pelletier [50] concluded that the calculation of minimum power input ($P_{min}$), for which the flight speed ($V_{mp}$) minimises energy cost per unit of time, was an appropriate model to estimate flight cost during non-migratory flights. This is supported by the similarity in predicted $V_{mp}$ estimated with Flight 1.24, and average flight speed measured empirically for Northern Gannets tracked using GPS loggers (15.2 m.s$^{-1}$ for $V_{mp}$ and 16.2 ± 0.5 m.s$^{-1}$ [42]). To estimate flight costs during migratory flights, we used the maximum range power ($P_{mr}$) calculated with the maximum range speed ($V_{mr}$); the latter is faster than $V_{mp}$, and is the speed at which the greatest distance is covered per unit of fuel energy consumption, thus minimizing the cost of transport [52]. This approach requires that flight muscle efficiency ($E_{fm}$), one of the parameters used in the model, is corrected for body mass, as $E_{fm}$ of birds performing forward flapping flight scales positively with mass [53]. We used the following equation to estimate $E_{fm}$: $E_{fm} = 0.3Mb^{0.137}$ where *Mb is* body mass. Since individual body mass was unknown during the non-breeding period and allometric scaling is based on an interspecific relationship in birds, we used the average body mass of instrumented birds during the breeding period (3.03 ± 0.3 kg). The other parameters were the average wing span (1.83 ± 0.04 m), wing area (0.262 m$^2$; [52]) and a ratio of flapping flight/gliding equal to 0.7 (proposed by Pennycuick [51] for a flap-gliding bird).

Estimates of flight costs 84 W ($P_{min}$) for foraging flights (winter season) and 103 W ($P_{mr}$) for long flights (fall and spring migrations) corresponding to $10 \times$ and $12 \times$ BMR, were calculated for gannets as a possible range of flight costs for this species. This used of two distinctive costs for flight during migration or winter is well supported by predictions from mechanical energetics pointed out by Pennycuick [54, 55] and Rayner [56].

To estimate thermoregulation costs per day, we used the method proposed by Garthe *et al.* [10] and summarized by the following equation: *thermal conductance* × (*core temperature–sea surface temperature*) × *time spent sitting on water per day* × *body mass*. The *thermal conductance* measured on two carcasses (in a swimming posture in a water tank) was 0.93 J.˚C$^{-1}$.g$^{-1}$.h$^{-1}$ where the *core temperature* of Northern Gannets was 39˚C [10].

**Statistical analysis.** Visualization of wintering locations was made using the packages 'ggmap' [48] and 'ggplot2' [49]. Timing of migration and breeding, time spent flying per day, sea surface temperature and energetics were compared between individuals with different wintering regions (GM, SE and NE) and between years (2010–2011 and 2011–2012) using linear mixed effects model (LMM, 'nlme' package in R) [57]. *Season*, *year* and *wintering area* were included as fixed factors and *individual* as a random factor [57]. Because sample sizes differed between migration categories, pair-wise comparisons were performed using least-squares means (LS means, '*lsmeans*' package in R) [58]. Hatching success was compared between migration categories and between years using loglinear models; three-way contingency tables where the response variable was the count (number of successful events) and the predictors were the categorical variables (*glm ()* function, '*stats*' package in R with a Poisson random component) [59]. Chi-squared tests were used to test for the interactions and pair-wise comparisons were performed using LS means. Conditions of normality and homogeneity of variances were tested graphically, with Shapiro–Wilk's normality test and Bartlett's test of homogeneity of variances. Variables that were not normally distribution were log-transformed. All statistical analyses were carried out using R software, version 3.5.3 [38] and results with $P < 0.05$ were considered significant. Values given in text are mean ± SE, and in graphs as mean ± 95% CI.

## Results

Tracking data were available for the complete non-breeding period for 58 Northern Gannets: 35 in 2010–2011 and 23 in 2011–2012, including 422,330 flights (7,282 ± 900 flights per individual). Deployment durations were 286 ± 67 days (including breeding period) and each individual was tracked for 192 ± 13 days during the non-breeding period. Of all individuals monitored, 12 spent the winter in the Gulf of Mexico (GM; 21%), 18 off the southeast coast of the U.S. (SE; 31%) and 28 off the northeast coast of the U.S. (NE; 48%) (**Fig 2**).

### Survival

The model $\varphi_t p_t$ was unequivocally the best supported by the resighting data. Only $t$ (year) had a significant effect on the annual survival rate. According to AICc, this model was a better fit than those including effects of both $g$ (migration strategy) and $t$ on survival (**Table 2**). Thus, wintering area had no influence on annual survival (lines faded on **Fig 3**). For the three first years of the monitoring, annual survival was estimated at 100% or so; thereafter, annual survival was fairly stable at around 84% (range: 79 to 90%). When we look at the survival rates separately for the three categories of migrants, we see that the rates are fairly similar, ranging from 69 to 100% (mean = 93%) for GM, 74 to 100% (mean = 88%) for NE, and 77 to 100% (mean = 90%) for SE.

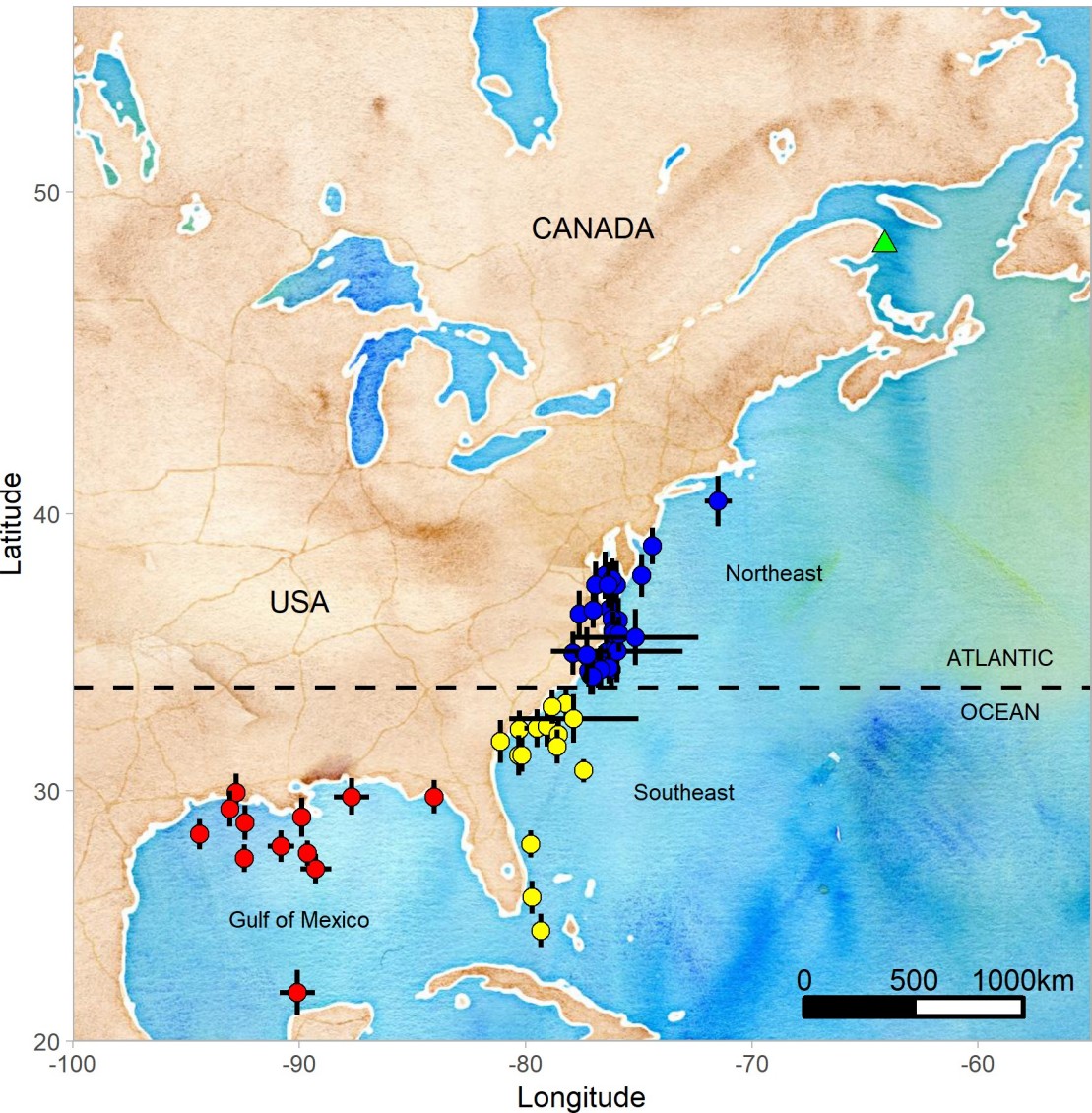

**Fig 2. Centroid of wintering location (Dec. 15th–Feb. 15th, mean ± 95% CI) of 58 Northern Gannets nesting on Bonaventure Island, Canada, equipped with a Global Location Sensor (GLS) logger and tracked in 2010–2011 or in 2011–2012.** Red dots presented individuals wintering in the Gulf of Mexico, yellow dots in southeast coast of the USA and blue dots in northeast coast of the USA. The map was generated in R (R Core Team, 2016) using the 'ggmap' [48] and 'ggplot2' packages [49] with map tiles by Stamen Design (www.stamen.com) and data by OpenStreetMap, under ODbL, under CC BY 3.0 (creativecommons.org/licenses/by/3.0/).

The probability of recapture ranged from 69% to 97% (mean: 86%). Recapture effort was lower at the beginning of the monitoring program (from 148 to 201 birds captured annually between 2009 and 2011), and nearly constant thereafter (285–330 gannets captured annually from 2012 onwards, which includes birds without GLS loggers). If we exclude the lowest probability of recapture, the mean rises by 3% and stays relatively constant between 81 and 97%.

## Phenology and hatching success

Northern Gannets arrived at their breeding grounds on Bonaventure Island on average on 20 April (± 5 days) with no difference between the three categories of migrants (**Table 3**, $F_{2,54}$ =

**Table 2. Models testing for differences in suvival probabilities of Northern Gannets nesting on Bonaventure Island, Canada, from 2009 to 2017.**

| Model | AICc | ΔAICc | $w_i$ | K | Deviance |
|---|---|---|---|---|---|
| ϕt pt | 421.28 | 0.00 | 0.95 | 15 | 166.60 |
| ϕg+t pt | 427.87 | 6.59 | 0.04 | 18 | 166.48 |
| ϕt p. | 429.74 | 8.47 | 0.01 | 9 | 188.10 |
| ϕg+t p. | 433.85 | 12.57 | 0.00 | 11 | 187.92 |
| ϕ. pt | 435.80 | 14.53 | 0.00 | 9 | 194.16 |
| ϕg pt | 439.94 | 18.66 | 0.00 | 11 | 194.01 |
| ϕ. P. | 442.93 | 21.65 | 0.00 | 2 | 215.85 |
| ϕg p. | 446.84 | 25.57 | 0.00 | 4 | 215.67 |
| ϕg*t pt | 451.78 | 30.51 | 0.00 | 31 | 159.63 |
| ϕg*t p. | 458.66 | 37.38 | 0.00 | 25 | 181.05 |

ϕ: survival probabilities; *p*: recapture probabilities; ΔAICc: difference between the model and the lowest AICc; $w_i$: AICc weight; **K**: number of parameters; g: categories of migrants (GM, SE, NE); **t**: time (years from 2009 to 2017);.: constant effect; *: interaction; +: additive effect.

0.9, $P > 0.05$). The mean post-breeding departure date of GM migrants was 3 October (± 15 days) (**Table 3**, $t_{16} = 3.19$, $P = 0.01$), 8.4 days on average before birds that migrated shorter distances (NE birds: 13 October [± 6 days]; SE birds: 10 October [9 days]). GM migrants spent 164 days at the colony during the breeding period, compared with around 175 days for birds that had migrated to the SE and NE.

Hatching success of Northern Gannets equipped with GLS loggers was between 45% and 100% between 2009 and 2017 (**Fig 4**). Loglinear models did not show an interaction between migrant category and hatching success ($\chi^2 = 0.52$, df = 2, $P = 0.77$). However, there was a significant interaction between year and hatching success ($\chi^2 = 33.46$, df = 7, $P < 0.0001$); pairwise comparison indicated that differences in hatching success were greatest between migrant categories in 2013 ($P = 0.0009$) and higher for GM and NE birds.

## Non-breeding period and flight behaviour

GM migrants spent about half as much time at their wintering area as NE and SE migrants, and about twice as much time on migration (**Fig 5A**, season [$F_{2,110} = 181.70$, $P < 0.0001$], wintering area [$F_{2,54} = 9.86$, $P < 0.0001$], season × wintering area [$F_{2,110} = 14.93$, $P < 0.0001$]). There was no significant difference between duration of the fall and spring migrations for all migrants ($P > 0.05$).

Whether they migrated to the GM, SE or NE, all birds spent 5.4 ± 0.8 hours flying per day during migration (**Fig 5B**, $F_{2,54} = 1.34$, $P > 0.05$) (corresponding to an average of 39 ± 4 flights per day of 8.8 ± 2.5 min per flight). However, as expected the time spent flying per day was lower in the winter when not migrating (3.8 ± 0.5 h on average) with a strong tendency to differ according to the wintering area ($F_{4,112} = 2.37$, $P = 0.06$). Gannets wintering in GM tended to spend 32 min less on average in flight per day than those wintering in SE and NE, but this was not statistically significant (compared to SE migrants, $t_{52} = -2.53$, $P = 0.24$; and compared to NE migrants, $t_{52} = -2.85$, $P = 0.13$).

## Sea surface temperature

Sea surface temperature around the breeding areas was 13 ± 1˚C. GM migrants were the only individuals that experienced warmer waters during winter than the breeding period (**Fig 6**).

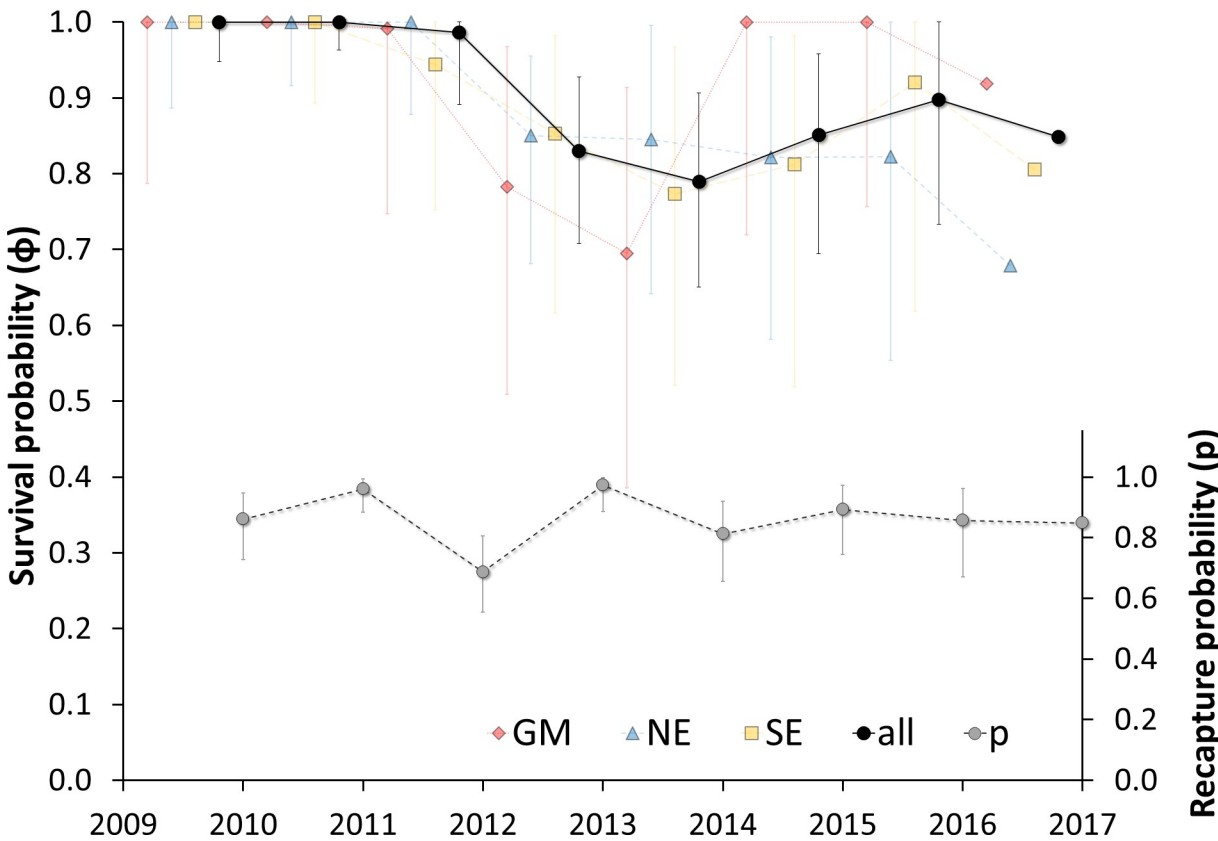

**Fig 3. Annual survival rates of Northern Gannets nesting on Bonaventure Island, Canada, and wintering in the Gulf of Mexico (GM), southeast coast of the U.S. (SE), northeast coast of the U.S. (NE) and all individuals together from 2009 to 2017.** Survival rates for GM, SE and NE were calculated from the model $\phi_{g*t} p_t$ and recapture probability (p) were calculated from the best fit model: $\phi_t p_t$ (see Table 2). Lines for sites are presented separately just to display the absence of difference, but they are faded because wintering sites are not in the best model (see Table 2). Bars are showing the 95% CI (estimated from profile likelihood method) around the mean annual survival.

From fall migration, sea surface temperatures began to differ in each area according to biological seasons ($F_{4,219} = 153.01$, $P < 0.0001$) and groups of migrants ($F_{2,54} = 44.86$, $P < 0.0001$). Gannets migrating to GM experienced waters that were 7˚C warmer than birds that wintered in NE ($16 \pm 1$˚C opposed to $9 \pm 2$˚C; $t_{51} = 10.67$, $P < 0.0001$), and during the spring migration back to the colony ($t_{51} = 9.16$, $P < 0.0001$).

## Energetics

Gannets expended twice as much energy flying per day during migration compared with that in wintering areas (**Fig 7A**, $F_{2,104} = 105.83$, $P < 0.0001$). Although there was a trend for GM

**Table 3. Arrival and departure dates (at or from the colony) of Northern Gannets nesting on Bonaventure Island, Canada, that wintered in the Gulf of Mexico (GM), southeast coast of the U.S. (SE) and northeast coast of the U.S. (NE) in winters 2010–2011 and 2011–2012.**

|  | GM migrants | SE migrants | NE migrants |
|---|---|---|---|
| *n* | 12 | 18 | 28 |
| Mean departure date (± SE, days) | 3 Oct (± 15)[a] | 10 Oct (± 9)[b] | 13 Oct (± 6)[b] |
| Mean arrival date (± SE, days) | 22 Apr (± 6)[a] | 20 Apr (± 5)[a] | 20 Apr (± 5)[a] |

[a, b]: different letters in two consecutive cells indicate significant difference ($p < 0.05$) between migrant categories

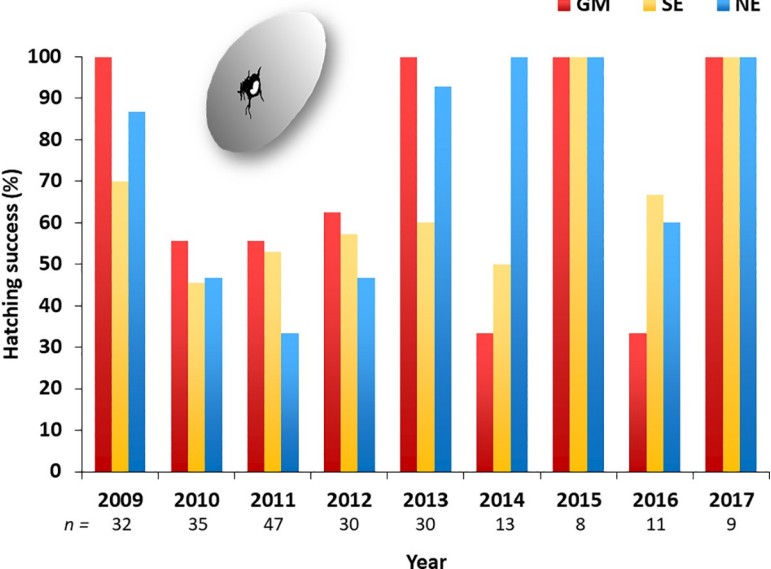

**Fig 4. Multi-year hatching success (%; number of chicks hatched/eggs laid) of three migrant groups of Northern Gannets nesting on Bonaventure Island, Canada, and wintering in the Gulf of Mexico (GM), southeast coast of the U.S. (SE) and northeast coast of the U.S. (NE) from 2009 to 2017.** There is no interaction between migrant groups and hatching success ($P > 0.05$) but an interaction between year and hatching success ($P < 0.0001$). Bottom line $n$ represent the total of nests for all categories. We used only the data for instrumented birds in 2010–2011 and 2011–2012, and for which we monitored hatching success between 2009 and 2017. Given the very high fidelity to wintering sites in gannets [25], we assume that individuals keep the same area year after year.

migrants to expend less energy flying in winter, the difference was not significant (season × wintering area: $F_{4,110} = 2.36$, $P = 0.06$). However, GM migrants spent less energy resting on water than NE and SE migrants during fall migration (**Fig 7B**, $t_{51} = -3.81$, $P = 0.01$), winter ($t_{51} = -9.58$, $P < 0.0001$) and spring migration ($t_{51} = -7.87$, $P < 0.0001$). When the energy spent flying and resting on the water was summed for each individual, energy expenditure of GM migrants was lower by 30% in winter than during the migration seasons. (**Fig 7C**, $t_{105} = 7.26$, $P < 0.0001$). When compared to NE migrants, GM migrants expended the least energy per day in the winter (-22%, $t_{51} = -5.46$, $P < 0.0001$) and spring migration (-11%, $t_{51} = -4.35$, $P = 0.002$). The differences for GM and SE migrants were also significant but smaller (**Fig 7C**).

If the total energy expended during the non-breeding period is estimated by multiplying daily energy expenditure by duration, there was no difference between the three categories of migrants (**Fig 8**, $F_{2,52} = 1.51$, $P = 0.23$).

## Discussion

Our study shows that the survival, hatching success, timing of breeding, and annual energy expenditure are similar between short and long-distance migrants in Northern Gannets. Despite substantial differences in wintering destinations and variation in flight behaviour (during migration and winter), there is no apparent carry-over effect of wintering location on the individual fitness in the subsequent breeding periods.

### GLS loggers, migration strategies and wintering areas

Recent studies of migrating birds have revealed wintering quarters of species for the first time, quantified the distance flown and the time spent moving, and often shown extensive variation

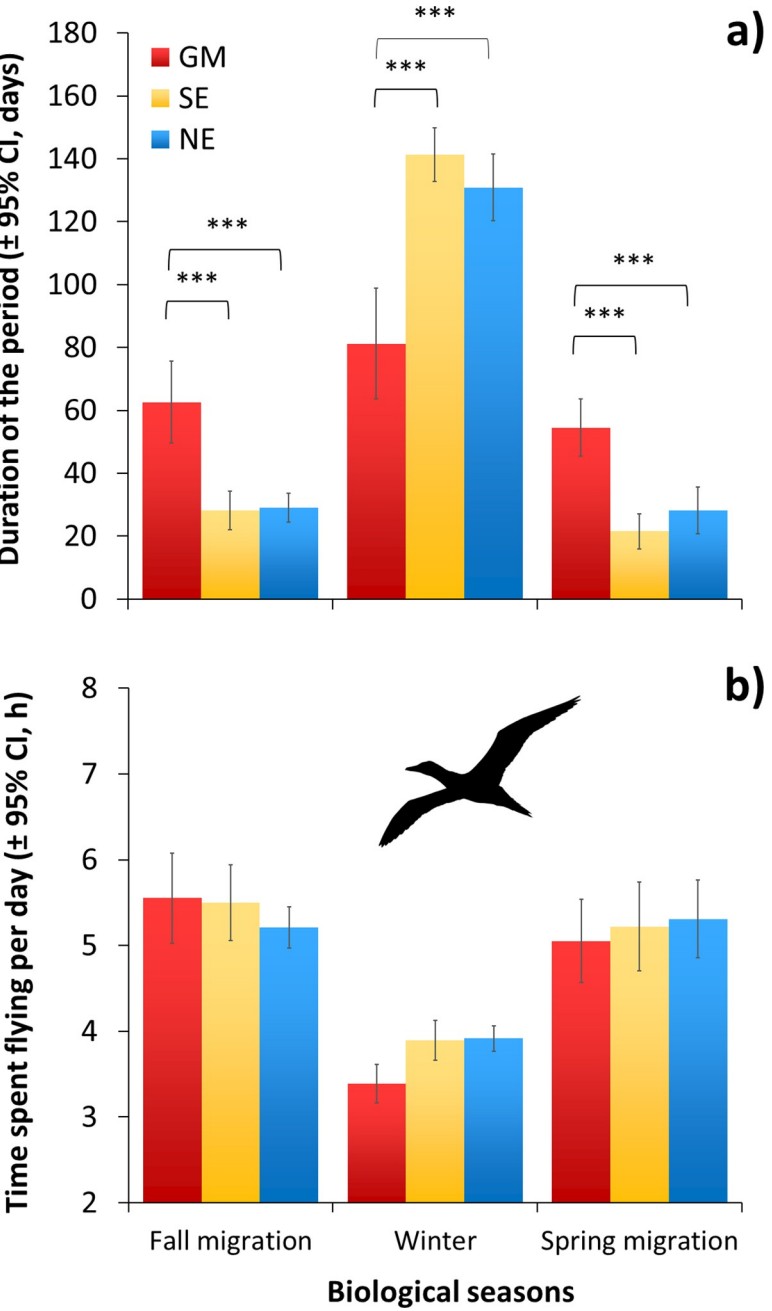

**Fig 5.** a) Duration of the non-breeding period (migrations and winter seasons) and b) time spent flying per day of three migrant groups of Northern Gannets nesting on Bonaventure Island, Canada. Wintering areas: Gulf of Mexico (GM, $n$ = 12), southeast coast of the U.S. (SE, $n$ = 18) and northeast coast of the U.S. (NE, $n$ = 28) in 2010–2011 and 2011–2012. (*: $P < 0.05$; **: $P \leq 0.01$; ***: $P \leq 0.001$).

among sexes [60], ages and individuals in migratory behaviour [61]. Recently, GLS loggers were used to study migration strategies of gannets nesting in the Northeast Atlantic [10, 23, 24, 26, 60] and Northwest Atlantic [25]. These loggers also made it possible to track long-term migration to assess marine pollution impact (e.g. Deepwater Horizon oil spill in the Gulf of Mexico in April 2010 [62–64]). In North America, these studies confirmed knowledge from

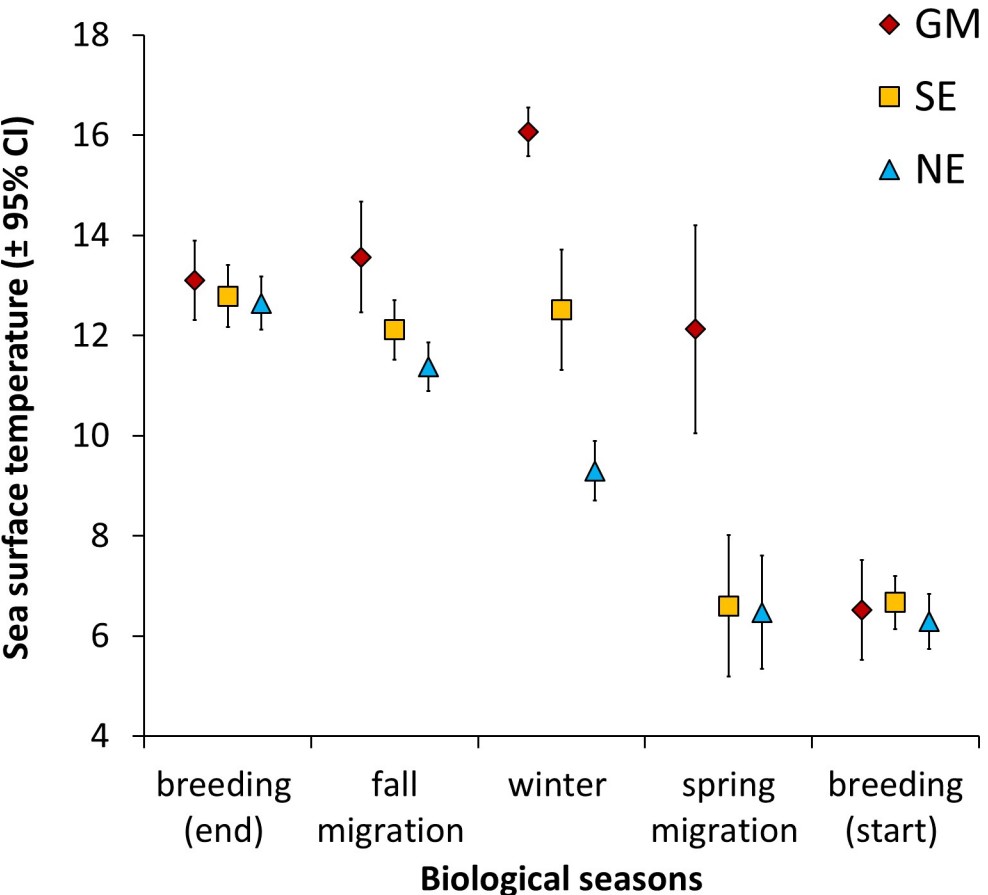

**Fig 6. Mean sea surface temperature (SST) recorded by loggers on Northern Gannets nesting on Bonaventure Island, Canada, throughout the annual cycle.** Wintering areas: Gulf of Mexico (GM, *n* = 12), southeast coast of the U. S. (SE, *n* = 18) and northeast coast of the U.S. (NE, *n* = 28) in 2010–2011 and 2011–2012.

banding and observations of birds at sea [46], that individuals vary in migration distance, spanning the area from the northern part of the American Atlantic coast to the Gulf of Mexico, but with no resident individuals in the Gulf of St. Lawrence, where waters are generally covered with ice during the winter. Across the breeding range of the species, migration strategies and non-breeding destinations are highly repeatable over consecutive years [10, 25, 26], and the net cost-benefit appears to be similar for all wintering areas in terms of energetics [10].

Gannets breeding on Bonaventure Island mostly dispersed along the Atlantic coast during the winter. In our study population, a larger portion were short-distance migrants either to the northeast coast (NE, 48%) or the southeast coast of the US (SE, 31%), and fewer (21%) were long-distance migrants to the Gulf of Mexico (GM) (**Fig 2**). The proportions of long-distance migrants from this colony are higher than those of ringing recoveries during the non-breeding period (6% [25, 62] or 8% [65]), but similar to wintering centroids calculated with previously collected geolocation data (24% [25] and 27% [62]). The main difference from both previous studies is the lower proportion of birds wintering in the SE (16% [25] and 11% [62])). Our geolocation results confirm that nearly a quarter of the adult gannets nesting at Bonaventure Island migrate to the GM, mainly around coastal Louisiana and Texas waters, with one exception near the Yucatan Peninsula. These results are similar to previous studies of Fifield *et al.* [25] and Montevecchi *et al.* [62]. In the NE area, the tracked gannets aggregated mainly in the

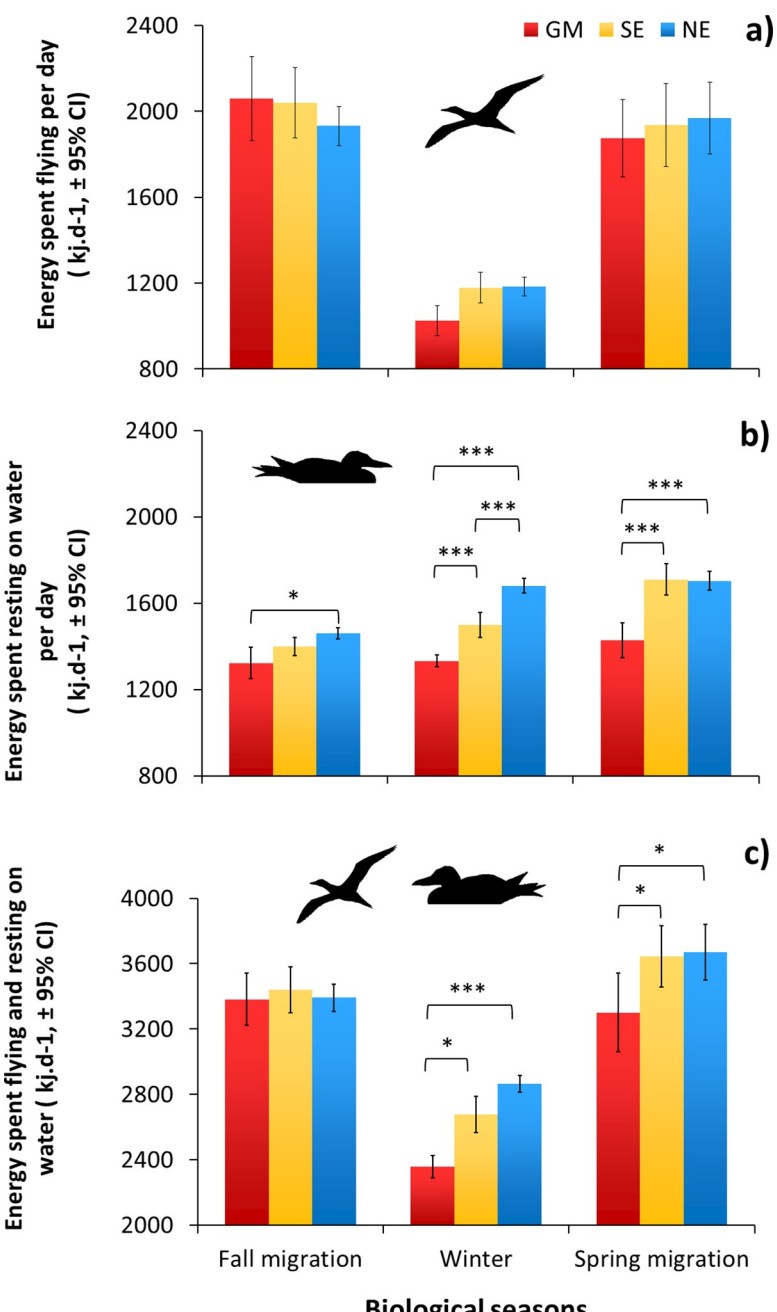

**Fig 7. Comparison of energy expended during the non-breeding period (fall migration, winter and spring migration) between three migrant groups of Northern Gannets nesting on Bonaventure Island.** Wintering areas: Gulf of Mexico (GM, $n$ = 12), southeast coast of the U.S. (SE, $n$ = 18) and northeast coast of the U.S. (NE, $n$ = 28) in 2010–2011 and 2011–2012. a) Energy spent flying per day, b) Energy spent resting on water per day, c) Energy spent flying and resting on water per day (*: $P < 0.05$; **: $P \leq 0.01$; ***: $P \leq 0.001$).

southern part of the zone around Delaware and Chesapeake Bays, and in the SE area, they frequented the coast of North Carolina, South Carolina and Georgia, with three individuals spending time off the Atlantic coast of southern Florida.

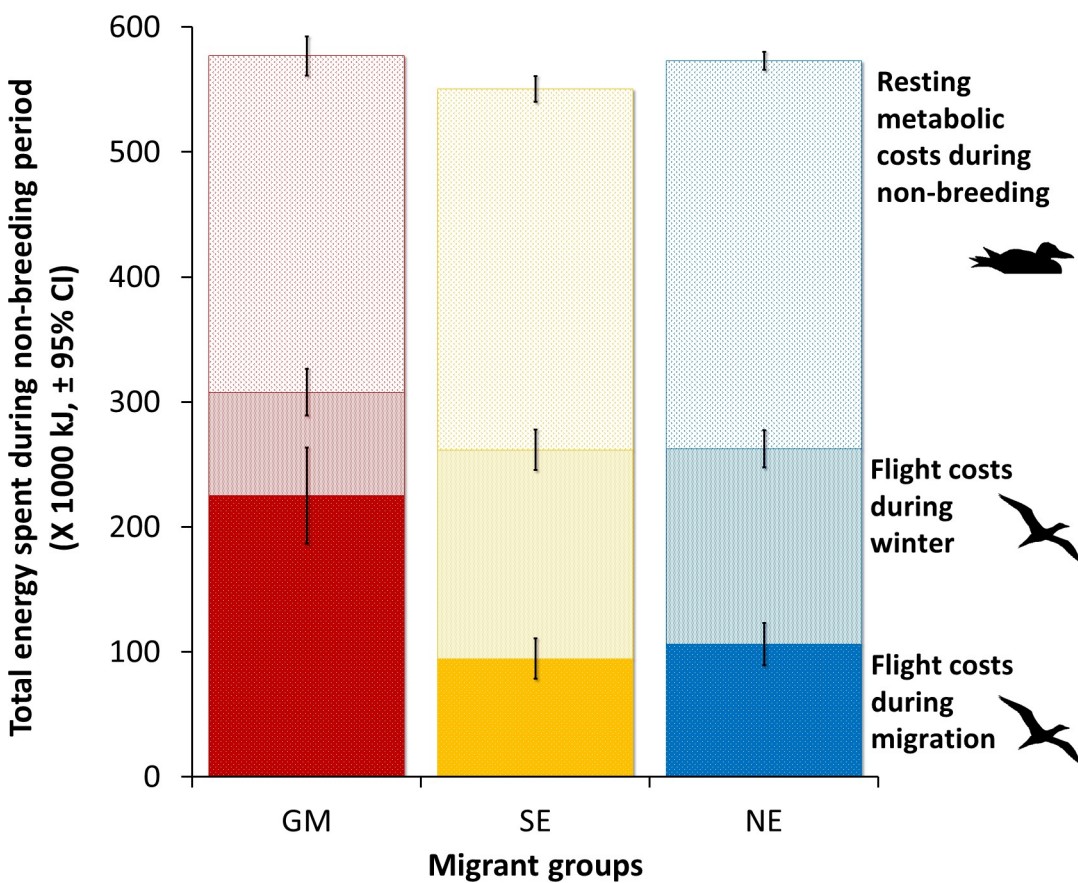

**Fig 8. Comparison of total energy spent during the non-breeding period (fall migration, winter and spring migration) between three migrant groups of Northern Gannets nesting on Bonaventure Island, Canada.** Wintering areas: Gulf of Mexico (GM, $n$ = 12), southeast coast of the U.S. (SE, $n$ = 18) and northeast coast of the U.S. (NE, $n$ = 28) in 2010–2011 and 2011–2012. Cumulative energy spent for resting metabolic were calculated by multiplying energy spent resting on water per day and duration of the non-breeding period. Cumulative energy spent flying during winter were calculated by multiplying energy spent flying per day during winter and duration of the winter season. Cumulative energy spent flying during migration were calculated by multiplying energy spent flying per day during migration and duration of both migrations (spring and fall).

## Survival

Our results indicate that there was no effect of wintering region on survival rate (**Fig 3**). Although confidence intervals for survival estimates were high because of the small sample size and the variability between years, the mean annual survival rate of 90% (pooling all wintering areas and years) was similar to the estimates from a previous study at different colonies (91.9%; [66]). At the species level, our results corroborate the literature where short-distance migrants species (or resident species) and long-distance migrants species show opposite trends in survival and reproductive rates: long-distance migrants tend to have higher annual survival than resident or short-distance migrants, but lower fecundity [number of broods and number of offspring produced per brood] [67, 68]). At the population level, survivorship is correlated positively with migration distance in some species (e.g. Mallards, *Anas platyrhynchos* [69]) but not in many others (e.g. Red Knots, *Calidris canutus* [70], many sparrows [71]). Newton [15] considered the merits of comparing populations of migratory versus resident bird species, and concluded that results from past comparisons were of dubious value, mostly because the populations differed in survival and reproduction for reasons other than just their migration

strategy (e.g. different breeding area). In our study, all the tracked birds migrated, and all were from the same colony, removing any confounding effect of breeding location on hatching success, survival, and recapture probability.

## Phenology and hatching success

In our study, hatching success from 2009 to 2017 (for all instrumented birds that we could monitor) was similar for birds with different wintering destinations, suggesting there is no clear carry-over effect in terms of a reproductive advantage from migrating shorter distances from the breeding colony. Further support for this conclusion is that the timing of arrival in spring was similar for all groups of migrants (**Table 3**), which is also the case for other gannet colonies in the western Atlantic (including Bonaventure) [25]. In contrast, Garthe *et al.* [10] found that the arrival date of gannets at the Bass Rock, UK, was later for individuals that wintered off west Africa (a migration of about 4000 km) than short-distance migrants, but not after sex was taken into account.

There is a certain synchrony in migration timing in gannets [25], and consequently, a small range of departure (fall) and arrival (spring) dates. Our results showed an overall range of 32 (2010) and 39 (2011) days for fall migration departure, and 19 (2010) and 18 (2011) days for spring migration arrival date. Date ranges and mean dates are consistent with data of Fifield *et al.* [25] for birds breeding in the same colony for arrival date (their study: 23 Apr ± 5) but different for fall departure (23 Oct ± 5). Fall departure is mainly determined to a large extent by the timing of chick fledging, through growth and development. This is corroborated by the characteristically earlier departure of failed breeders and nonbreeders than breeders [46], which is also observed in other seabird species (Cory's Shearwaters, *Calonectris borealis* [19], and Black-legged Kittiwakes, *Rissa tridactyla* [72]). The difference observed in mean departure date between short and long-distance migrants in our study could be explained by a lower fledging success for GM migrants in 2010 and in 2011. Unfortunately, because we were absent at the end of breeding period in the colony, we cannot know the exact fate of each chick. Alternatively, GM migrants may leave the colony before the other migrants solely because they anticipate a longer distance to be covered. Northern gannets are able to memorize the location of food patches and can at least to some extent anticipate the spatio-temporal distribution of prey [73], we may suppose that, as many other species, they are able on a circannual basis to synchronise their physiology and behaviour according to seasonal time cues for anticipating environmental periodicity, maximizing survival and reproductive success [74].

## Flight, thermoregulation and energy expenditure

The time spent flying per day was, as expected, lower during winter (3.8 h) than migration (5.4 h), regardless of migration destination. During winter, there was a tendency for birds to fly less per day (by 30 min) in the Gulf of Mexico, although this difference was borderline in terms of statistical significance ($P = 0.06$), probably due to the low sample size and substantial variability among individuals. Nevertheless, GM migrants would save about 25% in terms of overall foraging effort (through flight) compared with migrants to the other wintering areas, as observed in Atlantic puffins (*Fratercula arctica*) where flight behaviour and energy expenditure increase when birds winter further north [11]. Flight is the dominant behaviour during foraging in gannets, which search for prey in the air and then plunge-dive [75]. During migration, there were no significant differences in the time and energy spent flying per day between individuals travelling to different wintering areas. However, as expected, migrants took longer to reach the Gulf of Mexico than Atlantic destinations, leading to a greater overall cost of migration (**Fig 8**).

In contrast, energy expended while resting on the water was lower during both the migration and winter for gannets that migrated to the Gulf of Mexico than those wintering on the Atlantic coast (Fig 7B). This was related to the warmer SST in more southerly latitudes (Fig 6). Given that time spent on the water was the major part of the time budget, it represented a substantial daily energy cost, varying between 1300 and 1700 kJ per day. In comparison, flight costs were between 1000 and 1700 kJ per day (Fig 7A). Therefore, when flight costs are added to resting and thermoregulatory costs for the whole non-breeding period (migration and winter), there were no significant differences between strategies. On this basis, the energy costs of the alternative migration strategies observed in this population are likely to be broadly equal. However, our estimates of energy expenditure (EE) exclude the cost of swimming, diving and digestion, hence our knowledge of daily energy expenditure (DEE) is incomplete. Unfortunately, we are not aware of any estimate of DEE during the winter or migration seasons for gannets.

Our results can be compared with estimates of DEE during the chick-rearing period calculated with 'Seabird FMR Calculator' (https://ruthedunn.shinyapps.io/seabird_fmr_calculator/) [76] for Northern Gannet, and also with empirical estimates obtained for this species [77] and for a congener, the Cape Gannet (*Morus capensis*). With the 'Seabird FMR Calculator' (using body mass = 3,030 g, colony latitude = 48˚ and breeding phase = brood), the estimate of DEE is 1,343 $kJ.kg^{-1}.d^{-1}$ (lower confidence interval: 990 $kJ.kg^{-1}.d^{-1}$; upper confidence interval: 1,738 $kJ.d^{-1}.g^{-1}$). According to the empirical studies based on doubly labelled water, the estimate for Northern Gannet is 1,520 $kJ.kg^{-1}.d^{-1}$ [77] while two independent estimates for Cape Gannet are 1,478 $kJ.kg^{-1}.d^{-1}$ [78] and 1,374 $kJ.kg^{-1}.d^{-1}$ [79]. Regardless of the method, the values are essentially the same. Using the average of 1,429 $kJ.kg^{-1}.d^{-1}$ from these studies, a gannet of 3.03 kg departing from our colony in the fall would have a DEE of 4,329 $kJ.d^{-1}$. Our estimate of EE related to resting, thermoregulation and flight during the non-breeding period ranged between 2,400 and 3,600 $kJ.d^{-1}$, which represents about 55% to 83% of the total DEE of a breeding bird. During the winter, seabirds are no longer under the central-place foraging constraint, *i.e.* do not need to collect and return with food for the growing chick, and therefore show lower activity levels than during breeding [61, 80]. This decreasing activity level during the non-breeding period has been observed in many species: Procellariformes [27, 28, 30, 81], sulids [10], skuas [30] and alcids [29]. Therefore, we consider our estimate of EE to be a fair approximation of DEE, particularly as total energy expended while diving and for digestion are probably proportional to the time spent searching for prey during flight, as flying and diving activity are highly correlated in gannets ($r = 0.876$, $p < 0.0001$, [82]).

The similarity between annual energy expenditure of different group of migrants depends mainly of the total energy spent during the winter period (Fig 8). Our results suggest that the energetic benefits during the winter of foraging in the GM, probably associated with milder weather conditions (lower thermoregulatory costs), and greater prey abundance (requiring less time spent flying for foraging) outweigh any added costs of the longer migration. It is probably for these reasons that we did not detect any obvious carry-over effects.

## Conclusion

Carry-over effects have been demonstrated in many birds and mammals species [17] and seem to be common in migratory species [21]. Those effects occur when processes in the breeding period influence characteristics and success of the following non-breeding period, and vice-versa ([17] for a review). For example, unsuccessful breeders may be in better body condition at the end of the breeding period and, as their reproductive costs are lower, they may invest more time and energy for long-distance migration [72]. Conversely, favourable conditions

during winters may result in earlier arrival date on breeding grounds [83]. However, Bogdanova *et al*. [80] found no evidence of a link between choice of wintering grounds and breeding success. They hypothesised that their results could be influenced by differences in prevailing conditions, being poorer in the season preceding the winter. Indeed, carry-over effects can be modulated by environmental conditions, increasing when conditions are less favourable [84, 85]. Ramos *et al*. [86] did not detect clear effects of breeding effort on migratory behaviour or subsequent breeding attempts, suggesting that costs of breeding may be rapidly buffered during the non-breeding period by changes in other traits (*e. g*. moult). Life-history theory predicts that long-lived species should not invest too much in a single breeding event, since this may reduce their survival and future breeding prospects [87]. Carry-over effects should be more persistent and stronger in short-lived species [17] and may be apparent only in years with very poor environmental conditions [80, 86].

In conclusion, using a cost-benefit approach, we found no evidence of carry-over effects of wintering destination on fitness parameters (hatching success and adult survival) in gannets, nor did we find any differences in overall energy expenditure during the non-breeding period after accounting for time spent migrating. Overall, this suggests there are no obvious selective advantages to choosing a particular wintering region over another for this seabird species. However, further multi-year studies or experimental manipulations would better reveal the effects of environmental stochasticity during the breeding and non-breeding periods, and test for possible influences of other factors (e.g. moult schedule) on carry-over effects.

## Supporting information

**S1 Dataset. The final dataset used for statistical analysis, figures and tables is available in this file.** Appendix sheet includes m-array of the Northern Gannets of the three categories of migrants (GM: Gulf of Mexico, NE: northeast coast of the U.S., SE: southeast coast of the U.S.) marked and recaptured between 2009 and 2017. Number released on year *i* pools newly marked and previously marked individuals recaptured.
(XLSX)

## Acknowledgments

We would like to thank the volunteers and colleagues who participated in the field work on Bonaventure Island (Sylvain Christin, Isabeau Pratte, Marie-Anne Bergeron, Cynthia Franci, Mélanie Sabourin, Corentin Chaillon, Lynn Miller, several park naturalists, Nicolas Lepage, Jan Bouthillier, Roxane Bélanger) or for data analysis (Maude Valois-Bérubé, Noémie Martin-Chouinard, Gabrielle Robineau-Charrette, Catherine Babin and Mathieu Lévesque).

## Author Contributions

**Conceptualization:** David Pelletier, Magella Guillemette.

**Data curation:** David Pelletier.

**Formal analysis:** David Pelletier, Yannick Seyer, Salomé Bonnefoi.

**Funding acquisition:** David Pelletier, Magella Guillemette.

**Investigation:** David Pelletier, Magella Guillemette.

**Methodology:** David Pelletier, Magella Guillemette.

**Project administration:** Magella Guillemette.

**Resources:** David Pelletier, Richard A. Phillips.

**Software:** David Pelletier, Salomé Bonnefoi, Richard A. Phillips.

**Supervision:** David Pelletier, Magella Guillemette.

**Validation:** Magella Guillemette.

**Visualization:** David Pelletier.

**Writing – original draft:** David Pelletier, Yannick Seyer, Magella Guillemette.

**Writing – review & editing:** David Pelletier, Yannick Seyer, Stefan Garthe, Salomé Bonnefoi, Richard A. Phillips, Magella Guillemette.

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
