## [Decision Letter · Decision Letter 0]

2 Jan 2020

PONE-D-19-27127

So far, so good… Similar fitness consequences and overall energetic costs for short and long-distance migrants in a seabird

PLOS ONE

Dear Mr Pelletier,

Thank you for submitting your manuscript to PLOS ONE. After careful consideration, we feel that it has merit but does not fully meet PLOS ONE’s publication criteria as it currently stands. Therefore, we invite you to submit a revised version of the manuscript that addresses the points raised during the review process.

Firstly, I would like to apologise for the significant delay that this paper has undergone during the review process. This is not our normal practice, however as is the case sometimes, your paper hit a number of snags during the review process. Rest assured that these were procedural snags and not anything to do with the quality of your work.   

Your study has now been assessed by two reviewers, both of these reviewers found your paper to be interesting and have merit for publication. Reviewer 1 was very positive about the work and only had minor comments. Likewise, reviewer 2 was also very positive about your work, however, had a few more detailed comments that they felt needed addressing before the paper was suitable to publish. In particular please pay attention to their concern that some important models have not been included in the survival analysis. Please also ensure that the clarity of the text is improved through the manuscript. Once these concerns and the others raised by R1 and R2 have been addressed, then I see no further hurdles to publication.

We would appreciate receiving your revised manuscript by Feb 16 2020 11:59PM. To enhance the reproducibility of your results, we recommend that if applicable you deposit your laboratory protocols in protocols.io, where a protocol can be assigned its own identifier (DOI) such that it can be cited independently in the future. For instructions see: http://journals.plos.org/plosone/s/submission-guidelines#loc-laboratory-protocols

We look forward to receiving your revised manuscript.

Kind regards,

Andrew J. Hoskins

Academic Editor

PLOS ONE

Journal Requirements:

3. We note that Figure 2 in your submission contain [map/satellite] images which may be copyrighted. All PLOS content is published under the Creative Commons Attribution License (CC BY 4.0), which means that the manuscript, images, and Supporting Information files will be freely available online, and any third party is permitted to access, download, copy, distribute, and use these materials in any way, even commercially, with proper attribution. For these reasons, we cannot publish previously copyrighted maps or satellite images created using proprietary data, such as Google software (Google Maps, Street View, and Earth). For more information, see our copyright guidelines: http://journals.plos.org/plosone/s/licenses-and-copyright.

1. You may seek permission from the original copyright holder of Figure 2 to publish the content specifically under the CC BY 4.0 license. 

Reviewers' comments:

Reviewer's Responses to Questions

**Comments to the Author**

1. Is the manuscript technically sound, and do the data support the conclusions?

Reviewer #1: Yes

Reviewer #2: Partly

2. Has the statistical analysis been performed appropriately and rigorously? 

Reviewer #1: Yes

Reviewer #2: No

3. Have the authors made all data underlying the findings in their manuscript fully available?

Reviewer #1: Yes

Reviewer #2: No

4. Is the manuscript presented in an intelligible fashion and written in standard English?

Reviewer #1: Yes

Reviewer #2: No

5. Review Comments to the Author

Reviewer #1: This is a well presented and considered study. A good opening paragraph introduces the evolutionary trade-offs, and this follows throughout the paper. Results are well presented in figures and tables. The trade-off between additional flight costs to reach a more distant wintering site where thermoregulatory costs are lower makes sense, and so this study offers useful evidence.

The discussion is quite long. It includes valuable insight, however some general text could be removed or shortened so as to focus on the novelty of this study. For example, a brief further discussion could explore why these migration strategies persevere given the lack of carry-over effects in this population. Could it be to avoid competition at wintering sites, or because of familiarity with a location?

The methods are generally well presented. Flight costs are estimated using measurements of mass and wing size recorded in the field. By my understanding from the paper, this is calculated using average measurements from all birds (please clarify). Would it have been possible to calculate this for each individual, and if so would you expect much variation because of size differences between individuals? Perhaps worth clarifying in the text either way.

Please check grammar throughout. Some errors detailed below but there are others in text.

Line specific comments:

Title: I wonder whether the “So far, so good…” part of the title detracts from the clear and concise message in the rest of it. This is perhaps personal preference, though.

Line 24-25: Consider grammar of sentence. Is there a missing word? “Allows us …” perhaps? Or, “The study of differential migration offers unique opportunities to identify such mechanisms and compare the costs and benefits of migration.”

Line 35-36: This was not a significant result, consider removing as the abstract provides good evidence without it.

Line 69: Grammar: change ‘stays’ to ‘stay’

Line 138: The list of models here could benefit from addition introduction, including, for example, the description of parameters given in Table 2. The different models could be linked more specifically to the study hypotheses to aid readers, particularly those with less experience of survival analyses who are still interested in the broader reach of this study.

Line 159: Please clarify these components, here or in the text where model structure is introduced.

Paragraph commencing line 411: This is very broad, and the advances in tracking technology have been well discussed elsewhere. I therefore wonder whether some of this paragraph detracts from the impact and interesting discussion of the results of this study. Consider streamlining, and focussing on the novelty of this study.

Line 513: here, you compare DEE from two similar seabird species during the breeding season because of the absence of non-breeding information on DEE, which makes sense. However, to obtain an estimate of DEE during the breeding period for your specific population, you could use the ‘seabird FMR calculator’ proposed by Dunn et al. 2018: https://royalsocietypublishing.org/doi/full/10.1098/rsbl.2018.0190

Reviewer #2: This study aims to investigate the mechanisms underlying different migration strategies, using a differential migrant, the Northern Gannet, in the western Atlantic. The authors combine data on migration distance and phenology, thermal and locomotive energetics, survival and breeding success to identify the costs and benefits of short- and long-distance migrations. The authors found no statistically significant difference in the annual survival probability, timing of arrival at the breeding site or hatching success between birds undertaking short- or long-distance migrations. Although long-distance migrants are thought to experience higher energetic costs during migration (due to flying greater distances), reduced foraging effort and thermoregulation costs in the warmer wintering area appear to compensate for this to make overall energy expenditure equal between short- and long-distance migrants during the non-breeding period. The authors conclude that there are no apparent carry-over effects as a result of long-distance migration and that the costs and benefits of different migration strategies are equal for this species.

The study largely follows that by Garthe et al. (2012), which is a cost-benefit analysis of different migration strategies of Northern Gannets in the eastern Atlantic with similar findings. The authors have used a multi-faceted approach to present a convincing case for their conclusions and the study is an interesting and useful addition to the literature but I think there are a few things that need addressing before publication.

Major:

1. The authors tested six different models in the survival analysis (L138), but some apparently important models have been excluded. Survival has been modelled in relation to migration strategy, year and their interaction, but none of the models test whether survival is constant across all groups and years (i.e. . ., . t) or whether there is an additive effect of migration strategy and year (i.e. g+t ., g+t t).

2. L441: “The high survival rate for the first 3 years of the monitoring program is because we only included birds fitted with GLS that were re-captured.” This is confusing. Does this mean that individuals were only included in the initial marked and released cohort if they were later recaptured? If so, this 100% ‘survival’ rate is meaningless and these years should be removed from the analysis.

3. The m-array from the mark-recapture analysis should be made available in the supporting information so that the number of birds in each cohort is clear.

4. L483: “The difference observed in mean departure date between short and long-distance migration gannets could be explained with a lower fledging success for GM migrants. Unfortunately, because we were absent at the end of breeding period in the colony, we cannot know the exact fate of each chick. However, we can argue that this difference is unrelated with migration strategy and winter destination, but more associated with variations of local resources during breeding.” If the difference in departure dates is a result of lower fledging success for GM migrants, could this not be a carry-over effect? All birds are at the same colony so presumably have access to the same local resources, so a difference in fledging success between birds with different migration strategies may well be linked with those differences in migration, or to another factor that influences both migration strategy and resource availability/foraging success during breeding. To say that there is no effect on breeding success and no carry-over effects appears to be overstating the results.

5. L182: How did laying success of different migrant types compare? Were some individuals excluded from the analysis of breeding success because they did not lay eggs (i.e. failed to breed)?

6. The manuscript requires extensive additional proof-reading/copy-editing to improve readability and correct grammatical and typographical errors throughout. E.g. L24: “…and allows to compare the costs and benefits of migration…” should be reworded as, e.g. “…and allows comparison of the costs and benefits of migration…”

Minor:

1. L32: “Despite similar fitness consequences, we found, as expected, that the overall energetic cost of migration is higher for long-distance migrants, although the daily cost was similar between strategies. In contrast, daily maintenance and thermoregulation costs were lower for GM migrants…” This is confusing and the second sentence seems to contradict the first.

2. L35: ”…where sea-surface temperature was 4-7oC warmer.” Warmer where? In the wintering area?

3. L60: “Differential migration, broadly defined as observed variation in distance and destination…” Distance and destination? Or distance and timing?

4. L62: This sentence is unclear.

5. L64: “However, according to the review of Cristol et al., studies are ambiguous…” Are individual studies ambiguous? Or is it that relationships are difficult to untangle from the literature as a whole?

6. L77: I like the phrase “ballistic dispersal” but “dispersive migration” is perhaps more commonly used!

7. L92: “…that there is no long-term cost to migrate further south when winter areas provide benefits.” This is slightly confusing. The authors need to specify that the “winter areas” referred to are those further from the colony. As the sentence is referring to a “basic principle of differential migration”, the word “south” could be removed to make it more general.

8. L101: “Annually, around 180 nests…” This sentence seems irrelevant and misleading if these nests are not all included in the current study. From my understanding, 58 individuals were monitored for this study – those birds from which GLS devices were successfully retrieved and provided sufficient data.

9. L132: “We tested for permanent effect (heterogeneity in survival probability – transient effect) and temporary effect (heterogeneity in capture probability – trap-dependence).” I would suggest referring to these directly as transience and trap dependence instead of permanent and temporary effects as these terms are unclear. However, transience is not “heterogeneity in survival probability” and trap-dependence relates to heterogeneity in recapture probability. These definitions need rewriting.

10. L135: Provide test results as well as p-values here.

11. L137: Define the term ^ and state the ^ value.

12. L137: Provide a reference for the use of AICc.

13. L140: “The model was built to test the difference in survival among the three categories of migrants.” This sentence is confusing and could be deleted.

14. L158: “The CJS model is obtained…” The goodness of fit or χ2 value of the CJS model is obtained, rather than the model itself.

15. L163: …”the time spent in transit to and from the wintering area as the « fall migration season »…” Surely the fall and spring migration periods?

16. L192: What was the elevation angle that was used?

17. L199: What bandwidth (h-value) was used and why?

18. L219: This sentence is unclear.

19. L272: What is number of flights? This seems irrelevant to the current study.

20. L288: Specify that this refers to annual survival rates for birds using different wintering areas, rather than survival on the wintering areas. We already know that these survival rates are fairly similar because migration strategy was not included in the top model of survival.

21. L345: “There was no significant difference between duration of the fall and spring migrations for all migrants.” Is this referring to a lack of difference between duration of fall and spring migrations within each individual? If so, these should be identical because the duration of spring migration was assumed to be the same as duration of fall migration (see L210). If not, this sentence is unclear.

22. L380: “…was lower by 30% than during the migration seasons.” Lower when? During the winter period?

23. L383: State the effect size here.

24. L476: Specifying this study’s mean dates alongside those of Fifield et al. would aide comparison here.

25. L531: Preservation in what way?

26. L514: Define “DLW”.

27. References need proof-reading and formatting.

28. The study species could be included in the manuscript title, although I appreciate that this may be a personal preference.

29. The authors may wish to consider the following additional literature:

Amélineau, F., Péron, C., Lescroël, A. et al. (2014) Windscape and tortuosity shape the flight costs of northern gannets. Journal of Experimental Biology 217: 876-885, doi: 10.1242/jeb.097915

Deakin, Z., Hamer, K., Sherley, R. et al. (2019) Sex differences in migration and demography of a wide-ranging seabird, the northern gannet. Marine Ecology Progress Series 622: 191-201, doi: 10.3354/meps12986

6. PLOS authors have the option to publish the peer review history of their article (what does this mean?). If published, this will include your full peer review and any attached files.

Reviewer #1: No

Reviewer #2: No

---

## [Author Response · Author response to Decision Letter 0]

24 Feb 2020

Our detailed response to reviewers is in the file Pelletieretal_Response to Reviewers.pdf

---

## [Editor Report · Decision Letter 1]

26 Feb 2020

So far, so good… Similar fitness consequences and overall energetic costs for short and long-distance migrants in a seabird

PONE-D-19-27127R1

Dear Dr. Pelletier,

We are pleased to inform you that your manuscript has been judged scientifically suitable for publication and will be formally accepted for publication once it complies with all outstanding technical requirements.

With kind regards,

Andrew J. Hoskins

Academic Editor

PLOS ONE
---

## [Editor Report · Acceptance letter]

3 Mar 2020

PONE-D-19-27127R1 

So far, so good… Similar fitness consequences and overall energetic costs for short and long-distance migrants in a seabird 

Dear Dr. Pelletier:

I am pleased to inform you that your manuscript has been deemed suitable for publication in PLOS ONE. Congratulations! Your manuscript is now with our production department. 

With kind regards,

on behalf of

Dr. Andrew J. Hoskins 

Academic Editor

PLOS ONE